

# Effect of bioceramic intracanal medication on the dentinal bond strength of bioceramic cements: an *ex-vivo* study

Rahaf A. Almohareb[1], Reem M. Barakat[2], Fahda N. Algahtani[1], Mshael Ahmed Almohaimel[3], Denah Alaraj[3] and Norah Alotaibi[3]

[1] Department of Clinical Dental Sciences, College of Dentistry, Princess Nourah Bint Abdulrahman University, Riyadh, Saudi Arabia
[2] Dental Clinics Department, King Abdullah Bin Abdulaziz University Hospital, Princess Nourah Bint Abdulrahman University, Riyadh, Saudi Arabia
[3] College of Dentistry, Princess Nourah Bint Abdulrahman University, Riyadh, Saudi Arabia

## ABSTRACT

**Background:** The present study evaluated the effect of a bioceramic intracanal medicament (Bio-C Temp) on the push-out bond strength of bioceramic cements.
**Methods:** Forty-eight human single-canaled premolars were prepared and randomly divided into three groups: Group (A) received no intracanal medicament; Group (B) calcium hydroxide (CH); and Group (C) Bio-C Temp. After medicament removal, the roots were sectioned transversely. The slices in each group were separated into two subgroups ($n = 16$): in Subgroup (1), mineral trioxide aggregate (MTA) was placed, and in Subgroup (2) Bio-C Repair. Push-out bond strength was determined using a universal testing machine, applying a constant compressive force on the cement until bond failure. The failure mode was also evaluated. Data were analyzed using the Chi-square test and two-way ANOVA followed by Tukey's *post hoc* tests. The level of significance was set at 5%.
**Results:** The pushout bond strength of Bio-C Repair was significantly higher than that of MTA irrespective of intracanal medication ($p = 0.005$). The placement of Bio-C Temp was associated with significantly lower bond strength ($p = 0.002$, $p = 0.001$).
**Conclusion:** Bio-C Repair showed better bond strength compared to MTA, irrespective of intracanal medication. Bio-C Temp intracanal medicament, however, decreased the bond strength of both these cements.

## INTRODUCTION

Root canal therapy for teeth with wide open apices and incompletely formed roots presents some challenges (*Hachmeister et al., 2002*), such as performing effective root canal disinfection and achieving an efficient apical seal. Various treatment approaches have been proposed to manage such teeth. The traditional technique is multiple-visit apexification with intracanal calcium hydroxide ($Ca(OH)_2$) dressing, while a more contemporary method suggests using bioceramic material as an apical plug (*Pereira et al., 2021*).

Corresponding author
Reem M. Barakat,
rmbarakat@pnu.edu.sa

Eliminating microorganisms from root canal systems requires chemo-mechanical methods that involve using intracanal medicaments, such as $Ca(OH)_2$. More recently, calcium silicate-based intracanal medicaments have been proposed (*Alsubait et al., 2020*). One such material is Bio-C Temp (Angelus, Londrina, Brazil), a ready-to-use calcium silicate paste used as an intracanal dressing. Its low solubility in comparison to $Ca(OH)_2$ allows for a longer release of hydroxyl ions (OH), which increases the pH and inhibits bacterial proliferation.

An important factor affecting the clinical success of endodontic treatment is maintaining a sufficient seal and reducing the risk of obturation detachment, which depends on the bond strength between the root canal sealant and dentin (*Oktay, Ersahan & Gokyay, 2018*). However, intracanal medicaments, such as $Ca(OH)_2$, can alter the chemical structure and mineral content of dentin surfaces, interfering with the sealant-dentin bond (*Oktay, Ersahan & Gokyay, 2018*). The push-out bond strength test has been recommended to measure the adhesive strength of root-canal sealers because it is easy to reproduce and interpret and can provide a realistic assessment of the bond strength with dentin (*Oktay, Ersahan & Gokyay, 2018*).

Mineral trioxide aggregate (MTA) is the most common material used as an apical barrier in regenerative endodontic procedures and apexification. Excellent clinical results have been reported following the initial application of $Ca(OH)_2$ (*Moore, Howley & O'Connell, 2011*). MTA consists primarily of portland cement (75%), which includes tricalcium silicate, dicalcium silicate, tricalcium aluminate, and tetracalcium aluminoferrite. Bismuth oxide (20%) enhances the cement's radiopacity (*Camilleri, 2008*). Upon setting, the portland cement undergoes a hydration reaction, releasing $Ca(OH)_2$ and increasing the environmental pH, which promotes the formation of hydroxyapatite crystals (*Camilleri, 2008*). Despite MTA's good adhesion, it has several drawbacks, such as difficult manipulation, long setting times, tooth staining, and low resistance to washout before setting (*Oktay, Ersahan & Gokyay, 2018*).

Other calcium silicate-based materials with improved handling properties have been proposed as apexification/regeneration plug materials. Bio-C Repair (Angelus, Londrina, Brazil) is one such ready-to-use bioceramic cement. It is similar to white MTA in terms of cytotoxicity, biocompatibility, and biomineralization (*de Toubes et al., 2021*). Bio-C Repair also offers excellent adhesion to dentin and can be applied from a threaded syringe. It contains zirconia oxide as a radiopacifier, which has been shown to reduce tooth discoloration (*Kohli et al., 2015*).

Calcium silicate-based materials have recently been introduced for use as an intracanal medicament under the name Bio-C Temp. According to the manufacturer, Bio-C Temp contains polyethylene glycol, titanium oxide, a resin base, calcium tungsten as a radiopacifier, and an active component consisting of tricalcium aluminate, calcium oxide, and calcium silicate. It offers high biocompatibility, easy manipulation and removal, and high alkalinity (pH = 12) and radiopacity (*Bio-C Temp, 2022*).

Bio-C Temp has been reported to have a bioactive ability that supports the survival and differentiation of osteoblasts, promoting periapical repair (*Edanami et al., 2023*; *Lopes et al., 2024*). Some studies have examined Bio-C Temp's effects on dentin microhardness

(*Alshamrani et al., 2024*), sealer bond strength (*Escobar et al., 2023*; *Alshamrani et al., 2024*), and tooth discoloration (*de Campos et al., 2023*). However, there is a lack of information about its effects on the ability of calcium silicate-based cements to adhere to root canal dentin and their bond strength.

The push-out test is a standard method for evaluating bond strength. It measures how well a material resists dislodgement when a tensile load is applied parallel to the tooth's long axis. It helps assess the material's adhesive properties and, for a root canal filling material, determines its quality (*Collares et al., 2016*). This study aimed to evaluate the effects of Bio-C Temp placement on the push-out bond strength of Bio-C Repair compared to that of MTA. The null hypothesis was that Bio-C Temp would not affect the bond strength of Bio-C Repair material and MTA.

## MATERIALS AND METHODS

This study received an exemption approval from the Internal Review Board at Princess Nourah bint Abdulrahman University (IRB no. 22-0235). Sound human premolars with a single, straight root canal, extracted for periodontal or orthodontic reasons after obtaining patient consent, were randomly selected. The inclusion criteria were teeth with single root canals, minor apical curvature (<5°), and free from calcification, internal or external resorption, previous endodontic treatment, caries, or cracks. Teeth with excessively short roots (<20 mm) were excluded. The external root surfaces were scaled with ultrasonic instruments and washed with distilled water to remove any calculus or soft tissue. The sample size calculation was conducted using G*Power 3.1 software (Heinrich-Heine-Universität, Düsseldorf, Germany), considering a power of 90% and a type 1 error probability of 5%. Assuming an effect size of f = 0.5, the required sample size was determined to be 48 teeth.

### Teeth preparation

According to the majority of studies conducted on push-out bond strength (*Brichko, Burrow & Parashos, 2018*), the following tooth preparation and subsequent push-out bond test was performed: Crowns were sectioned below the cemento-enamel junction to achieve a root length of 16 mm for all roots. The root canals were shaped with ProTaper Universal (Dentsply International, Tulsa, OK, USA) rotary files up to size F5. Using a 27-gauge needle (Medic, Jiangsu, China) 5 mL of 2.5 percent sodium hypochlorite (NaOCl) (Pharma Vitality, Riyadh, Saudi Arabia) was used to irrigate the canals between each file size. Pesso reamers sizes 1–5 were consecutively used to construct a standardized parallel canal space with a diameter of 1.25 mm and a length of 10 mm. To remove the smear layer, the canals were irrigated with 5 mL of 17% ethylenediaminetetraacetic acid (EDTA) (PD EDTA 17%, Produits Dentaires SA., Vevey, Switzerland). A final 10 mL of irrigation with saline was used to halt any extended effects of the previous irrigants. The canals were dried with article cones (Meta Biomed, Chungcheongbuk-do, Republic of Korea).

The teeth were randomly allocated to one of three groups based on the intracanal medicament: a control group (Group A) where no medicament was placed; $Ca(OH)_2$ paste (UltraCal™ XS, Ultradent, Cologne, Germany) was placed in Group B canals; and Bio-C

Temp (Angelus Indústria de Produtos Odontológicos, Londrina-PR, Brazil) was placed in Group C. The intracanal material was introduced into the canals using its respective syringe carrier. The applicator tip was inserted 3 mm short of the canal's working length, then gradually withdrawn as the canal was filled. To confirm that the canal was completely obturated, mesio-distal and bucco-lingual radiographs were taken for each tooth. The access cavity was then sealed with a temporary filling material simulating the *in vivo* situation, and the specimens were placed in an incubator at 37 °C/100 percent humidity for 3 weeks. Following the 3 weeks, the canals were re-opened and gently irrigated to remove the intracanal medicament with 5 mL 2.5% NaOCl and 10 mL 17% EDTA (PD EDTA 17%, Produits Dentaires SA., Vevey, Switzerland), alternating with 10 mL saline. All canals were dried with article points (Meta Biomed, Chungcheongbuk-do, Republic of Korea). All tooth preparation procedures were performed by a single operator experienced in root canal treatment.

The teeth were embedded in a cylindrical silicon mold measuring 12 mm in diameter. Using a Buehler Isomet low-speed saw (BUEHLER, Lake Bluff, IL, USA) with continuous water irrigation, the middle third of each root was sliced into two 2.0 ± 0.05 mm parallel transverse sections ($n = 96$ slices) in the coronal-to-apical direction. Randomly, specimen of each group were further separated into two subgroups ($n = 16$) according to the type of calcium silicate cement placed: Subgroup (1) MTA (MTA Angelus White, Angelus Indústria de Produtos Odontológicos, Londrina-PR, Brazil) prepared according to manufacturer instructions, and Subgroup (2) Bio-C Repair (Angelus Indústria de Produtos Odontológicos, Londrina-PR, Brazil), which comes in a pre-mixed, ready-to-use syringe. The cement was applied by a single operator within the lumens of the slices and condensed on a flat glass plate using an endodontic plugger (5/7, HuFriedyGroup, Lake Bluff, Illinois, USA). Finally, the specimens were wrapped in distilled water-soaked gauze and preserved in an incubator at 37 °C for one week (*VanderWeele, Schwartz & Beeson, 2006*).

## Push-out bond strength testing

The push-out bond strength was determined by an operator blinded to the groups, using a universal testing machine (Instron testing machine, Model 5967; ITW, Danvers, MA, USA). Each disk was mounted on a secure support jig with a 4 mm hole in the center to enable the plunger to move freely. A constant compressive force at a speed of 1 mm/min was applied using a 1.2 mm stainless-steel push-out rod placed in contact with only the calcium silicate cement material until the entire bond failed. The maximum force at material displacement was registered. Bond strength was calculated according to the following formula:

Bond strength (MPa) = force for displacement (N)/bonded surface area (mm$^2$).

Bonded surface area = 3.14 × radius of root canal × slice thickness.

A blinded operator examined the slices under a digital microscope (KH-7700, Hirox Co., Tokyo, Japan) at a magnification of 50X to determine the mechanism of failure. Failure was classified into three types: adhesive failure at the dentin-material contact, cohesive failure that occurs within the material, and mixed failure, which is a mixture of the previous two.

## Statistical analysis

The data were analyzed using SPSS software (version 22; SPSS Inc., Chicago, IL, USA). The data were checked for normality using the Shapiro–Wilk test, and accordingly, a two-way analysis of variance (ANOVA) followed by Tukey's *post hoc* multiple comparison test was performed. An independent samples *t*-test was used to compare the push-out bond strengths of the two bioceramic cements. A chi-square test was used to determine whether there was a significant association between the type of failure and the plug material or intracanal medicament. A significance level of 5% was used.

## RESULTS

Irrespective of the intracanal medication used, the push-out bond strength was significantly higher with Bio-C Repair ($52.47 \pm 58.05$ MPa) than with MTA ($21.86 \pm 45.05$ MPa; $p = 0.005$). The mean and standard deviation of the push-out bond strengths of MTA and Bio-C Repair in the different groups are shown in Table 1 and Fig. 1. Bio-C Repair had the highest push-out bond strength after placement of $Ca(OH)_2$ as an intracanal medicament ($88.29 \pm 69.99$). However, there was no significant difference in bond strength between the $Ca(OH)_2$ group and the control group ($p = 0.37$). In contrast, bond strength was significantly lower when Bio-C Temp was used as the intracanal medicament compared to $Ca(OH)_2$ and no medicament ($p < 0.001$ and $p = 0.001$, respectively).

Three types of material failure were observed: adhesive, cohesive, and their mixture (Fig. 2). The type of failure was significantly associated with the cement and intracanal medicament ($p = 0.002$ and $p < 0.001$, respectively). The percentage for each material failure type in each group is shown in Table 2. While MTA was associated with adhesive failure (68%) when placed in canals previously treated with Bio-C Temp, Bio-C Repair was associated with cohesive failure (56%). Regardless of intracanal medicament, MTA failure was predominantly mixed (48%), whereas Bio-C Repair failure was predominantly cohesive (Fig. 3).

## DISCUSSION

The endodontic filling material must adhere to dentin to effectively obturate the canal space and reduce the risk of endodontic treatment failure. A strong adhesive bond to dentin can enhance tooth fracture resistance and improve clinical longevity for endodontically treated teeth (*Komabayashi et al., 2020*). Strong adhesion is particularly important for teeth with open apices to prevent the filling material from dislodging during post-and-core placement or due to functional forces. The present study used the push-out test as an effective technique to determine how well calcium silicate-based root filling material adheres to dentin after placement of a calcium silicate-based intracanal medicament (*Jain et al., 2019*). This test is considered nontechnique sensitive, easy to perform, and can mimic certain clinical conditions (*Dem et al., 2019*). In addition, care was taken in specimen selection and preparation to attain accurate standardization.

Based on the current findings, Bio-C Repair demonstrated greater bonding strengths to dentin than MTA, irrespective of intracanal medication. This result is consistent with previous studies (*Nagas et al., 2016*; *Prado et al., 2021*) comparing premixed calcium

**Table 1 Mean and standard deviation for the push-out bond strengths of MTA and Bio-C Repair in the different experimental groups.**

| Bioceramic cement | Intracanal medicament | Mean ± Std deviation | 95% Confidence interval | | p value |
|---|---|---|---|---|---|
| | | | Upper limit | Lower limit | |
| MTA | Control (no medicament) | 29.50 ± 37.71[a] | 9.41 | 49.60 | $p < 0.0001$ |
| | Ca(OH)$_2$ | 31.45 ± 66.45[a] | −3.96 | 66.86 | $p = 0.001$ |
| | Bio-C Temp | 4.43 ± 2.85[b] | 1.01 | 8.28 | |
| Bio-C Repair | Control (no medicament) | 64.67 ± 40.76[c] | 42.97 | 86.41 | |
| | Ca(OH)$_2$ | 88.29 ± 69.99[c] | 50.99 | 125.59 | |
| | Bio-C Temp | 4.64 ± 6.81[b] | 2.90 | 5.95 | |

Notes:
Different lowercase letters indicate the presence of significant differences between the values.
$p$ value set at ≤0.05.
MTA, Mineral trioxide aggregate.

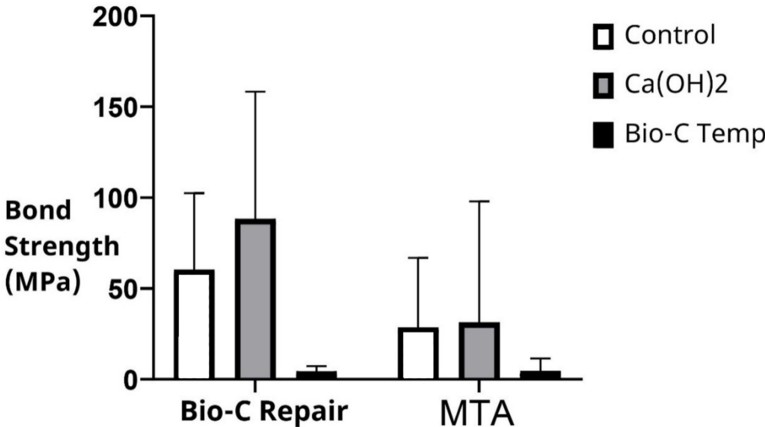

**Figure 1 Bond strength of Bio-C Repair and MTA following different intracanal medication.** MTA, Mineral trioxide aggregate.

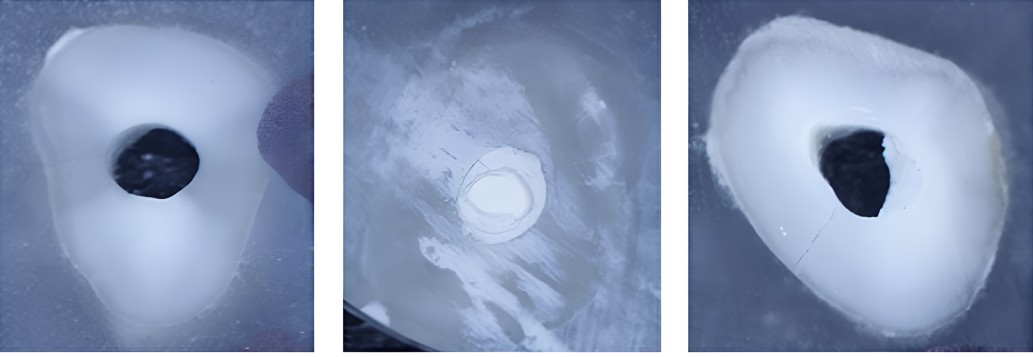

**Figure 2 (A) Adhesive failure of MTA after placement of Bio-C Temp intracanal medicament, (B) Cohesive failure of Bio-C Repair following placement of Ca(OH)$_2$, and (C) mixed failure type of MTA placed in canal previously treated with Ca(OH)$_2$.**

**Table 2 The percentage of failure types in each experimental group.**

| Intracanal medicament | Cement | Failure type | | |
|---|---|---|---|---|
| | | Adhesive | Cohesive | Mixed |
| No medicament (control) | Bio-C Repair | 1 (6.2%) | 8 (50%) | 7 (43.8%) |
| | MTA | 1 (6.2%) | 2 (12.5%) | 13 (81.25%) |
| Ca(OH)$_2$ | Bio-C Repair | 2 (12.5%) | 8 (50%) | 6 (37.5%) |
| | MTA | 7 (43.8%) | 1 (6.2%) | 8 (50%) |
| Bio-C Temp | Bio-C Repair | 3 (18.8%) | 9 (56.2%) | 4 (25%) |
| | MTA | 11 (68.8%) | 3 (18.75%) | 2 (12.5%) |

**Note:**
MTA, Mineral trioxide aggregate.

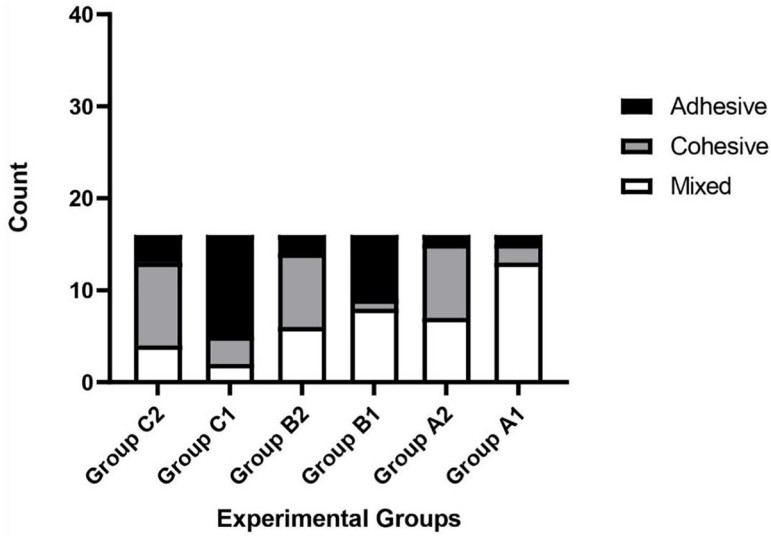

**Figure 3 Frequency of failure types observed in the different experimental groups.** A1, No intracanal medicament with MTA; A2, No intracanal medicament with Bio-C Repair; B1, Ca(OH)$_2$ paste with MTA; B2, Ca(OH)$_2$ paste with Bio-C Repair; C1, Bio-C Temp with MTA; C2, Bio-C Temp with Bio-C Repair.

silicate cements with conventional MTA. One explanation is the smaller particle size of Bio-C Repair (~2 microns), which promotes deeper penetration into the dentinal tubules, stimulating crystal growth and increasing its micromechanical retention (*Nagas et al., 2016*). Bioceramic cements containing nanoparticles also showed significantly lower porosity (*Yalniz et al., 2024*). Using a chelating irrigant to remove the intracanal medicament may have improved Bio-C Repair's adhesion to dentin by opening the dentinal tubules and facilitating the penetration of its smaller particles into them (*Kadić et al., 2018*; *Sfeir et al., 2023*).

The presence of zirconium oxide in Bio-C Repair may also contribute to its superior bonding strength. Zirconium oxide forms a strong 3D network by creating CaZrPO4 and increases free calcium during Bio-C Repair hydration reaction (*Prado et al., 2021*). Differences in preparation and handling properties between MTA and Bio-C Repair could

also be influential. Bio-C Repair comes in ready-to-use premixed syringes, while MTA powder needs to be mixed manually (*Prado et al., 2021*).

The amount of bioactive elements released by various bioceramic materials when they come into contact with dentin fluid varies. These elements enhance chemical bonding to the dentin and do not necessarily match the manufacturers' claims (*Badawy & Abdallah, 2022*).

The placement of $Ca(OH)_2$ as an intracanal medicament is crucial because it exhibits antibacterial properties and promotes the formation of hard tissue. These beneficial effects stem from its ability to break down into calcium ions ($Ca^{2+}$) and hydroxide ions ($OH^-$). Calcium hydroxide pastes, designed for immediate use, are formulated with specific vehicles that control the rate of this breakdown, ensuring a sustained release over time (*Fava & Saunders, 1999*).

In the present study, the application of $Ca(OH)_2$ resulted in the highest mean push-out bond strength values for both MTA and Bio-C Repair. However, it was not statistically significant compared to the control groups in which no intracanal medicament was placed. This aligns with previous studies that reported no difference in the bond strength of fast-setting, pre-mixed calcium silicate cements with prior placement of $Ca(OH)_2$ (*Alsubait et al., 2020*; *Gokturk & Ozkocak, 2022*). Other studies have reported an improvement in the bond strength of calcium silicate cements with prior $Ca(OH)_2$ application (*Nagas et al., 2016*; *Al-Haddad et al., 2020*), possibly due to the cements' reaction with $Ca(OH)_2$ residue, which improved their marginal adaptation. By contrast, $Ca(OH)_2$ was found to act as a barrier that impairs MTA adaptation and reduces its bonding strength (*Ghabraei et al., 2017*; *Alsubait et al., 2020*).

Bio-C Temp is an intracanal medicament that exhibits an elevated release of Ca2+ ions (*Oliveira et al., 2021*) and increases dentin microhardness (*Alshamrani et al., 2024*). Previous studies showed that it increased the bond strength of sealers such as AH Plus (*Alshamrani et al., 2024*). However, in the present study, it was associated with the lowest bonding strength values. A possible reason for this reduction in bond strength is the remaining remnants of Bio-C Temp. Although there is limited information on the efficacy of Bio-C Temp removal from the root canal system, it is well known that current techniques cannot completely eliminate intracanal medicaments (*Wigler et al., 2017*; *Zhou, Liu & Guo, 2021*). Essentially, Bio-C Temp, Bio-C Repair, and MTA are calcium silicate-based. According to the manufacturer, polymeric chains present in Bio-C Temp hinder the connectivity of the hydrated calcium silicate particles, preventing them from hardening. It seems that the residue of this paste may interact with calcium silicate-based filling materials and interfere with their setting or hinder the formation of chemical bonds between cement and dentin, thus impairing its adaptation to dentin and lowering the push-out bond strength (*Villa et al., 2020*). Bio-C Temp has also been reported to cause tooth discoloration (*de Campos et al., 2023*) and be cytotoxic to human dental pulp cells (*Oliveira et al., 2021*).

In the present study, MTA was frequently associated with adhesive or mixed failures, irrespective of the intracanal medicament placed. This observation indicates that MTA's internal cohesion surpassed its adhesion to the dental surface, which could be attributed to

its large particle size, which hinders its penetration into the dentinal tubule (*Aly, El Shershaby & El-Sherif, 2020*). In contrast, Bio-C Repair was associated with cohesive failures, especially after the placement of Bio-C Temp. This finding is consistent with previous studies on TotalFill Root Repair Material (FKG Dentaire, La Chaux-de-Fonds, Switzerland), which is similar to Bio-C Repair (*Kadić et al., 2018*; *Alsubait et al., 2020*).

Previous studies have reported conflicting results regarding the failure type of MTA. Some have reported adhesive failures (*Saghiri et al., 2013*; *Dawood et al., 2015*), while others reported mixed failures (*Marques et al., 2018*; *Alsubait et al., 2020*). The reasons for these discrepancies include possible differences in material storage conditions, variations in dentinal tubule number and size within the slices, and differences in methodology, such as sectioning the roots after placement of the cement materials.

A major limitation of this study was its inability to replicate various clinical conditions *ex vivo*, such as the contamination of materials with blood during the setting reaction. Long-term clinical trials are needed to establish the clinical implications of these findings. In addition, micro-computed tomography assessment of gap formation was not performed before push-out bond strength testing. This omission could have obscured the impact of material adaptation on the study findings. Moreover, the placement of bioceramic cement plugs after root sectioning does not faithfully replicate the clinical scenario and may have influenced the results (*Brito-Júnior et al., 2015*).

Considering the study limitations, Bio-C Repair appears superior to MTA in bond strength making it a recommended choice for use as an apical plug in teeth with open apices.

$Ca(OH)_2$ showed no significant impact on the bond strength between dentin and bioceramic cement supporting its use as a disinfecting intracanal medicament in cases of wide apices. However, Bio-C Temp resulted in a decrease in interfacial bond strength. Further studies are needed to understand the reasons behind the reduced bond strength observed with Bio-C Temp. Additionally, more research is recommended to explore how different irrigation materials, protocols, and cement application techniques affect the push-out bond strength of bioceramic cements.

## CONCLUSIONS

The null hypothesis was rejected. Bio-C Repair showed better bond strength than MTA, irrespective of the intracanal medicament. Bio-C Temp decreased the push-out bond strength of calcium silicate-based cements, while the placement of $Ca(OH)_2$ did not adversely affect the bond strength of these cements.

### Funding

This research was funded by Princess Nourah bint Abdulrahman University Researchers Supporting Project number (PNURSP2024R162), Princess Nourah bint Abdulrahman University, Riyadh, Saudi Arabia. The funders had no role in study design, data collection and analysis, decision to publish, or preparation of the manuscript.

### Grant Disclosures

The following grant information was disclosed by the authors:
Princess Nourah Bint Abdulrahman University, Riyadh, Saudi Arabia: PNURSP2024R162.

### Competing Interests

The authors declare that they have no competing interests.

### Author Contributions

- Rahaf A. Almohareb conceived and designed the experiments, analyzed the data, prepared figures and/or tables, authored or reviewed drafts of the article, and approved the final draft.
- Reem M. Barakat conceived and designed the experiments, performed the experiments, analyzed the data, prepared figures and/or tables, authored or reviewed drafts of the article, and approved the final draft.
- Fahda N. Algahtani performed the experiments, authored or reviewed drafts of the article, and approved the final draft.
- Mshael Ahmed Almohaimel performed the experiments, authored or reviewed drafts of the article, and approved the final draft.
- Denah Alaraj performed the experiments, authored or reviewed drafts of the article, and approved the final draft.
- Norah Alotaibi performed the experiments, authored or reviewed drafts of the article, and approved the final draft.

### Ethics

The following information was supplied relating to ethical approvals (*i.e.*, approving body and any reference numbers):

The study received an approval exemption, according to the Internal Review Board at Princess Nourah bint Abdulrahman University (IRB no. 22-0235).

### Data Availability

The datasets generated and analyzed in this study are available in the Supplemental File.

### Supplemental Information

Supplemental information for this article can be found online at http://dx.doi.org/10.7717/peerj.17826#supplemental-information.

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
