# Peer review of "Effect of bioceramic intracanal medication on the dentinal bond strength of bioceramic cements: an ex-vivo study"

_PeerJ, doi:10.7717/peerj.17826_

## Round 0.1 · original submission · Major Revisions

Dear Authors,

Thank you for your submission to PeerJ. The manuscript requires some revisions and responses to the following comments/questions:

I noticed a lack of information about the BIO-C® TEMP intracanal medication in the introduction section. Most of the introduction focuses on MTA. Please provide a more detailed introduction to the medication. Additionally, in the discussion section, please elaborate on the clinical applicability of your study.

Throughout the methods section, include the manufacturer information for all materials and equipment used.

Lastly, please consider having the manuscript polished for English language and grammar.

Reviewer 1 ·

Basic reporting

The basic reporting is fine. However Authors cited a few old literature which is outdated.
Dental material science is changing very fast. Therefore please do not cite references older than 10 years.
Authors omitted a few important and latest articles related to the topic e.g
Evaluation and comparison of the porosity of different bioceramic-based materials, using micro-computed tomography (micro-CT) doi:10.17219/dmp/146663
Evaluation of the chemical composition and elemental distribution of 4 RCSs (1 resin-based and 3 bioceramic-based) by using energy dispersive X-ray spectroscopy (EDX), field emission scanning electron microscopy (FE-SEM) and elemental mapping after root canal obturation, both coronally and apically doi:10.17219/dmp/133954
The systematic review which aimed to compare and evaluate the effect of resin-based sealers and bioceramic sealers on postoperative pain after endodontic treatment doi:10.17219/dmp/155885
I suggest to present above mentioned papers.

Experimental design

Authors have to report which international guidelines (ISO or ADA) or maybe other sources have been used for teeth preparation and Push-out Bond Strength Test performance.

Validity of the findings

I assess the usefulness and validity of the findings as high. I don't have remarks regarding validity.

Additional comments

Authors have to present clear limitation of the study at the end of Discussion.
Authors have to add a legend of used abbreviations below each table and figure.
Please move this sentence "According to the results of this study, the null hypothesis was rejected." to the Conclusions (this is conclusion).

·

Basic reporting

Clarity and Language:
- The article is well-written, with clear and concise language.
- The title and abstract are informative and reflect the study's content accurately.

Literature and Background:
- The introduction provides a good background on the topic, explaining the importance of dentinal bond strength in endodontic treatments.
- References are relevant and up-to-date, but the introduction could benefit from a brief review of previous studies to better contextualize the current research.

Structure and Presentation:
- The article follows a logical structure, with distinct sections for the introduction, methods, results, and discussion.

Experimental design

Study Design:
- The ex-vivo design is appropriate for the research question, allowing controlled conditions and precise measurements.


Methodology:
- The sample size, inclusion and exclusion criteria, and randomization process are not sufficiently detailed. This information is crucial to assess the study's rigor and potential selection bias.
- The description of the intervention (bioceramic intracanal medication) and the control groups needs to be more detailed, including specifics on the materials and procedures used.
- The methods for measuring bond strength should be clearly described, including the type of test (e.g., push-out test) and how it was conducted to ensure reliability and validity.
Statistical Analysis:
- The statistical methods used should be explicitly stated, including the specific tests applied to compare groups. Reporting of confidence intervals and p-values is essential for assessing the significance of the findings.

Validity of the findings

Internal Validity:
- The study should detail any blinding procedures used to minimize bias, particularly in the measurement of outcomes.
- Potential confounding factors should be acknowledged and addressed.

External Validity:
- The applicability of the findings to clinical practice should be discussed. The sample should be representative of the broader population of teeth typically encountered in clinical settings.


Results Interpretation:
- The results should be interpreted with consideration of both statistical significance and clinical relevance. Any observed differences should be discussed in terms of their practical implications for endodontic treatments.

Additional comments

Strengths:
- The study addresses a relevant and important question in endodontics.
- The use of an ex-vivo design allows for controlled and precise measurements.

Areas for Improvement:
- More detailed reporting on the methodology, including randomization, sample selection, and measurement techniques.
- A more thorough discussion of potential biases and limitations.
- Clearer presentation of results, including all necessary statistical details.
- Strengthen the discussion by linking findings to existing literature and exploring their clinical implications.
- Future Directions: Suggestions for future research should be included, particularly studies that could address the limitations of the current study or explore related questions in clinical settings.

---

## Round 0.2 · accepted · Accept

Dear authors
Thank you for submitting your article to PeerJ. We are pleased to inform you that it is now deemed worthy of publication.

Reviewer 1 ·

Basic reporting

The manuscript has been correctly revised. I don't have further comments.

Experimental design

The manuscript has been correctly revised. I don't have further comments.

Validity of the findings

The manuscript has been correctly revised. I don't have further comments.

Additional comments

The manuscript has been correctly revised. I don't have further comments.

·

Basic reporting

No comment

Experimental design

No comment

Validity of the findings

No comment

Additional comments

No comment